# Engaging primary care professionals in suicide prevention: A qualitative study

**Elke Elzinga**[1]*, **Anja J. T. C. M. de Kruif**[2], **Derek P. de Beurs**[3], **Aartjan T. F. Beekman**[4,5], **Gerdien Franx**[6], **Renske Gilissen**[1]

**1** Research department, 113 Suicide prevention, Amsterdam, The Netherlands, **2** Department of Health Sciences, VU University Amsterdam, Amsterdam, The Netherlands, **3** Department of epidemiology, Netherlands Institute of Mental Health and Addiction (Trimbos Institute), Utrecht, The Netherlands, **4** Department of Research & Innovation, GGZ inGeest Specialized Mental Health Care, Amsterdam, The Netherlands, **5** Psychiatry, Amsterdam Public Health (research institute), Amsterdam UMC VU University, Amsterdam, The Netherlands, **6** Department of implementation, 113 Suicide prevention, Amsterdam, The Netherlands

* e.elzinga@113.nl

**Data Availability Statement:** Data cannot be shared publicly because the interviewees have not given consent for this, and the raw interview transcripts may contain sensitive or identifiable information. The transcripts are archived on a

## Abstract

In health systems with strongly developed primary care, such as in the Netherlands, effectively engaging primary care professionals (PCPs) in suicide prevention is a key strategy. As part of the national Suicide Prevention Action Network (SUPRANET), a program was offered to PCPs in six regions in the Netherlands in 2017–2018 to more effectively engage them in suicide prevention. This implementation study aimed to evaluate to what extent SUPRANET was helpful in supporting PCPs to apply suicide prevention practices. From March to May 2018, 21 semi-structured interviews have been carried out with PCPs and other non-clinical professionals from SUPRANET regions in the Netherlands. Verbatim transcripts were analysed using the grounded theory approach. Data was structured using the Consolidated Framework for Implementation Research, which enabled identifying facilitating and challenging factors for PCPs to carry out suicide prevention practices. An important challenge included difficulties in assessing suicide risk (intervention characteristics) due to PCPs' self-perceived incompetence, burdensomeness of suicide and limited time and heavy workload of PCPs. Another important limitation was collaboration with mental health care (outer setting), whereas mental health nurses (inner setting) and SUPRANET (implementation process) were facilitating factors for applying suicide prevention practices. With regard to SUPRANET, especially the training was positively evaluated by PCPs. PCPs expressed a strong need for improving collaboration with specialized mental health care, which was not provided by SUPRANET. Educating PCPs on suicide prevention seems beneficial, but is not sufficient to improve care for suicidal patients. Effective suicide prevention also requires improved liaison between mental health services and primary care, and should therefore be the focus of future suicide prevention strategies aimed at primary care.

secured server of 113 Suicide Prevention. Anonymized excerpts of transcripts are available upon request for researchers who meet the criteria for access to confidential data. Requests can be made via email to the corresponding author (Elke Elzinga; e.elzinga@113.nl) or to the research department of 113 Suicide Prevention (onderzoek@113.nl).

**Funding:** This study was funded by the Ministry of Health of the Netherlands: grant number 326961, 2017. The funders had no role in study design, data collection and analysis, decision to publish, or preparation of the manuscript.

**Competing interests:** The authors have declared that no competing interests exist.

## Introduction

Suicide is a major health issue. The WHO estimates that 800,000 people worldwide die as a result of suicide every year, which comes down to 2,192 suicides every day [1]. Suicide is a complex phenomenon in which social, cultural and biological factors interact [2]. Therefore, multilevel suicide prevention approaches are preferred above single, standalone measures [3–6].

The European Alliance Against Depression (EAAD) is an example of a multilevel approach. It was founded in 2004 with the purpose of creating a network of countries that have implemented action-focused, community-based interventions to treat depression and prevent suicides [7]. This approach was first tested in 2000 in a region in Germany (Nuremberg), where the total number of suicidal acts decreased by 24% compared to a control region [8]. The model has since been implemented in over 115 regions worldwide [9]. The rationale of the EAAD is that the various levels, including primary care, general public, community facilitators and high-risk groups, interact to create a synergistic and catalytic effect [10]. In the Netherlands, the model focuses on suicide prevention alone and is therefore named Suicide Prevention Action NETwork (SUPRANET). SUPRANET was initiated by 113 Suicide Prevention, the national suicide prevention centre, as part of the national agenda for suicide prevention commissioned by the Ministry of Health [11].

Although all levels of EAAD are relevant for suicide prevention, specific attention is given to primary care. Primary Care Professionals (PCPs) include both General Practitioners (GPs) and their Mental Health Support Staff (MHSS); the latter is a relatively new profession and refers to professionals who offer therapy sessions to primary care patients with mental health, psychosocial or psychosomatic complaints [11]. GPs are often in contact with patients shortly before they engage in suicidal behaviour [12–14]. Additionally, in many health care systems, among which the Netherlands', GPs function as gatekeepers to identify and refer suicidal patients [15]. Within SUPRANET, a program was developed to support PCPs carrying out evidence-based suicide prevention practices. It contains among others a training to increase their ability to explore and detect suicidal feelings and they are encouraged to improve continuity of care by enhancing collaboration with specialised Mental Health Care (MHC) and other health or community care organisations.

Supporting primary care is among the most effective suicide prevention strategies [5, 16, 17]. However, implementation of interventions in primary care is challenging [9, 18, 19]. An often-used framework to address implementation challenges is the Consolidated Framework for Implementation Research (CFIR). This comprehensive framework describes factors that are important in implementing and evaluating complex interventions. It consists of five domains (characteristics of the intervention, outer setting, inner setting, characteristics of individuals, and process of implementation) which interact and determine effectiveness of implementation together [20]. Using a qualitative design, we conducted this implementation study to evaluate to what extent SUPRANET was helpful in supporting PCPs to apply suicide prevention practices. These insights will be used to engage PCPs more effectively in suicide prevention by improving the use of SUPRANET.

## Methods

### Design

Semi-structured interviews based on a topic list were used to gather the perceptions and experiences of PCPs with regard to SUPRANET. The CFIR model was used to structure and organize the data. In the present study, *intervention characteristics* refers to PCPs' experiences with

applying suicide prevention practices such as exploring suicidal feelings. *Outer setting* refers to the level of collaboration with MHC and in *inner setting* we discuss the role of MHSS with regard to suicide prevention. The *process of implementation* describes how the implementation strategy, SUPRANET, was received by PCPs.

## SUPRANET for PCPs

In 2017, SUPRANET was implemented in six pilot regions in the Netherlands. The suicide prevention training was offered to both GPs and MHSS between June 2017 and May 2018. In total, 67 PCPs (49 GPs and 18 MHSS) from 23 GP practices completed the suicide prevention training before May 2018 (the study deadline). PCPs were additionally prompted to work on continuity of care by strengthening their collaboration with social community teams, emergency rooms, and MHC services [11]. Additional materials, including the module on suicide and medication, the suicide prevention guide, flyers and posters and a checklist, were provided to the PCPs (see Table 1).

## Study participants

We used convenience sampling and the snowball technique to recruit study participants from the PCPs within the SUPRANET regions who completed the suicide prevention training. We reached out to PCPs directly or to contact persons in GP practices via email and by phone. In regions where implementation was lagging, i.e., where no PCPs were being trained before the study deadline, professionals who were involved with the implementation of SUPRANET were approached for an interview. In total, 18 PCPs and three non-clinical professionals (two project leaders and one trainer) participated in this study. Financial compensation was available for PCPs who took part in SUPRANET.

## Data collection

A topic list was used to guide the semi-structured interviews (See S1 Material). This instrument consists of a list of topics and prompts based on literature [21–23] and discussed within

**Table 1. Various elements of SUPRANET for PCPs.**

| Component | Content |
|---|---|
| Suicide prevention training | The training consists of a theoretical section covering the epidemiology, suicide behaviour, and process of suicide, the Chronological Assessment of Suicidal Events (CASE) methodology, and treatment and referral. The tuition is interspersed with interactive exercises focused on connecting with feelings of despair, involving relatives, and diagnosing. The training is provided by experienced trainers from the Dutch College of General Practitioners and is accredited for four hours. |
| Continuity of care | Improving collaboration with social community teams, emergency rooms, crisis and MHC services by organizing meetings to discuss issues and make agreements about the treatment policy for suicidal patients. |
| Module on suicide and medication | A two-hour (accredited) individual e-learning or a group-based pharmaceutical therapeutic audit session about the role of medication in suicide prevention. |
| Suicide prevention guide | Contains a summary of the most important information from the suicide prevention training and the multidisciplinary guideline for the diagnosis and treatment of suicidal behaviour, and includes a triage tool to assist in referring suicidal patients. |
| Flyers and posters | Flyers and posters are aimed at patients and relatives, to encourage them to talk about and seek help for suicidal feelings. |
| Checklist | The checklist can be completed after consultations with (possible) suicidal patients. It includes items such as assessment of suicidal feelings, rumination, and the concreteness of suicidal plans. |

the research group (AK, DB, RG, EE). The rationale for this approach is that it is best suited to capture the perceptions and experiences of participants [24] and it leaves space for participants talk freely and bring up topics or issues themselves. The topic list was compiled to explore PCPs' perceptions and experiences with SUPRANET, suicide assessment and management in general and their views about liaison with other services. After the first couple of interviews were conducted, we reviewed the topic list and made some minor changes. Instead of focusing on the specific elements of SUPRANET, we focused on SUPRANET as a whole.

Interviews started after the participants had given consent for audio-taping and transcribing the interviews verbatim. Anonymity was guaranteed throughout the entire process. The interviews were conducted by one researcher (EE), who had completed certified training in qualitative research methods including interview skills, in the period October 2018 to March 2019. Sixteen interviews were held in a face-to-face setting. The remaining five interviews were, for logistical reasons, conducted over the phone. The interviews lasted from 21 to 64 minutes.

### Data analysis

The transcripts were analysed using the Grounded theory approach. This approach offers a systematic and rigorous process of data collection and analysis, thus facilitating in-depth study of phenomena with the aim of constructing theory from the data [25, 26]. Because of the inductive code structure that evolves from this process, it is most suitable for reflecting the experiences of participants [27]. The Grounded theory approach includes a step-by-step coding procedure, consisting of open coding, axial coding, selective coding reduction, and integration. During the open coding phase all transcripts were read thoroughly by two researchers (EE and a research assistant) and codes were allocated intuitively. Disagreements were discussed until consensus was reached. Then, codes were clustered into themes and categories in a thematic map. Eventually, codes and themes were fitted again onto the transcripts. Finally, during selective coding, underlying differences and similarities were studied, explanations were sought, and literature was integrated. The process of analysis was supported by an experienced qualitative researcher (AK) to ensure validity and reliability. MAXQDA software was used for data analysis [28]. The consolidated criteria for reporting qualitative research (COREQ) was used as checklist for explicit and comprehensive reporting of qualitative data [29].

### Ethical approval

According to the Dutch Medical Research Involving Human Subjects Act, this kind of observational study is exempt from ethical review, since the participants were not subjected to any procedure nor were they asked to follow rules of behaviour. Nevertheless, we followed standard ethical procedures by obtaining informed consent from the participants and guaranteeing confidentiality and anonymity throughout the research process.

### Results

In this qualitative evaluation study, we interviewed in total 21 participants of whom 13 (62%) were GPs, five (24%) were MHSS and three (14%) were non-clinical professionals involved with the implementation of SUPRANET ("Other") (see Table 2).

**Table 2. Characteristics of the participants.**

|  | Male | Female | Total |
|---|---|---|---|
| **GP** | 8 | 5 | 13 |
| **MHSS** | 2 | 3 | 5 |
| **Other** | 2 | 1 | 3 |
| **Total** | 12 | 9 | 21 |

GP = General Practitioner, MHSS = Mental Health Support Staff

## Intervention characteristics

**Barriers with regard to assessing suicide risk in the general practice.** Many PCPs brought up issues concerning the complexity and unpredictability of suicidality when exploring the issue with patients. They argued that suicidal ideation is difficult to recognize, that it exists in various forms, and that a lot of factors are involved in the suicidal process. PCPs also reported about patients who they felt use suicidality as a strategy to attract attention or as a means of manipulation, making it more challenging to differentiate between actual suicidal patients and patients who may need other forms of care. Additionally, some patients reportedly felt inhibited about discussing mental issues and, therefore, whether consciously or not, presented to their GP with physical complaints. PCPs found it hard to discuss suicidal feelings with these patients straight away and argued that it could take multiple consultations before crossing that bridge.

PCPs said that most patients did not give out distinctive signals indicating suicidality, nor did they proactively disclose their suicidal feelings. Therefore, they argued, it is necessary to actively explore suicide risk. However, many found it hard to determine when to explore this. Most PCPs reported they did not follow a systematic approach for assessing suicidal feelings, but relied on their gut feeling or on the course of the consultation when deciding to explore these thoughts. All PCPs agreed with the recommendation of the multidisciplinary guideline for diagnosis and treatment of suicidal behaviour, which states that suicidal feelings should be assessed during every depression-related consultation, although some argued that it is difficult to apply in everyday practice.

PCPs reported that patients' suicidal behaviour strongly impacts their professional life. PCPs who experienced one or multiple fatal suicides in their career stated that they carried this experience with them for the rest of their life, especially if they felt that they did not handle the situation properly, which may lead to feelings of self-doubt and failure.

*"I have lost someone who was in care in an MHC institution but who had also visited me just before. She jumped from a building [. . .]. I felt like maybe I should have. . . even if she had come to me about her little toe, maybe I should have kept asking further. Asking her how she was really doing. I failed in that." (R20)*

Although PCPs often sympathized with their patients when they experienced suicidal thoughts, they could also be perceived as a burden, especially when patients were in suicidal crisis and could not be left alone.

*"I once had an experience where a person thought this was the hospital. This man was so suicidal that I just stayed with him until the psychiatrist arrived in the evening. In that case, you get sort of stuck with the patient. I thought, if I let him walk out, he'll jump straight out in front of a car." (R4)*

To address the suicidality of patients can impose a heavy burden on PCPs, who can feel pressured, stressed, and sometimes even personally responsible for their patient's life.

Of the PCPs, especially GPs reported an enormous workload and very limited time. Not only for consultations with patients but also for extra activities in their GP practice, such as taking part in training- or research. It also played a role in the assessment of suicidal thoughts during consultations:

*"When someone in the waiting room has chest pain and a suicidal or depressed patient who has an appointment with an MHC specialist tomorrow consults me, I decide not to ask any further. This might not be good for this patient, but overall it might be for the best."* (R20)

However, most PCPs did not agree with this statement and argued that patients who consult during surgery hours are generally stable and that medical assistants will alert them when this changes. Besides, there are more options than either discussing the matter right away or not discussing it at all. Most PCPs argued they prioritized their patient's mental wellbeing over their schedule.

*"A couple of times, it has happened that consultations take a bit longer, which is really annoying if it's scheduled for 10 minutes. But in the end, these patients say, 'Thank you, thank you for taking the time.' And then, those 10 minutes extra are suddenly a great gift." (R18)*

## Outer setting

**Collaboration with MHC institutions.**   Although it varied per region, many PCPs reported issues with regard to collaborating with MHC services. They argued that MHC services operate in isolation and fail to communicate information about patients. Another frequently mentioned issue was difficulties with access to MHC:

*"Sometimes you have to pull some serious strings to get someone reviewed or referred. That is difficult. Access is difficult." (R16)*

This resistance was especially noticeable when patients were not in severe crisis, but when the PCPs were worried anyway and required support.

Further, many PCPs reported that processes are very time consuming. Speaking to a professional on the phone takes a long time, admitting a patient into care, starting treatment, or the provision of feedback are inordinately time consuming. While PCPs feel, especially when they refer a patient to the crisis service, that the situation is urgent and requires a fast response.

Additionally, PCPs expressed a need for transparency regarding the methods and procedures in MHC. Some stated that they did not know what happens with a patient during treatment in MHC.

*"If you send patients to MHC, I sometimes say, 'They disappear into a black box. That box is shaken a couple of times and at some point, they fall out.' I never know what happens to my patients in the meantime, except maybe that they have been relocated six times. [. . .] Sometimes I question whether–with all due respect–crisis services provide better care than I do." (R2)*

Another concern, which is also hinted at by the PCP from the quote above, is the quality of care. Patients are being transferred frequently and see many different health care professionals,

who do not always follow the treatment plan that was agreed upon or sometimes not even showing up for an appointment. This kind of behaviour may evoke feelings of rejection and feeling rejected by the very organization that is supposed to provide help, hinders recovery even more.

*"Crisis services arouse the feeling that patients experienced before: rejection. They already feel rejected and then they get it all over again." (R9)*

PCPs found the service provision of MHC institutions was substandard. They expressed the desire to collaborate more closely, including having the opportunity to call and discuss patients or to send them in for review, while patients remained treated in primary care. They want to work together, not to shuffle patients back and forth.

*"It would be very helpful if they would welcome us in a friendly way, not just the patients, but us too as fellow caregivers. Because we can get at ease from these conversations, 'Have you already thought of this or that?'. That can make a big difference. Often, we don't need crisis services to take over but just to provide backup." (R9)*

Most PCPs were very understanding about the situation of the MHC services, which had suffered budget cuts, staff shortages, and onerous policy changes. Nevertheless, they were unsatisfied with the current collaboration and insisted on improvement. Most PCPs reflected on their own practice as well, arguing that they could also improve, but that this too required collaboration and communication.

## Inner setting

**Role of the MHSS.**   GPs reported that the MHSS was an enormous relief in terms of providing care for suicidal patients. MHSS are trained to support patients with mental health problems and they have more time than GPs to spend with patients, whereas GPs cannot always provide the close monitoring that these patients may require. Some GPs even argued that MHSS should be given more responsibilities with regard to suicide prevention.

*"I think you should make people aware that MHSS play an important role in this; they may even be more important than GPs. They often have more time, expertise, and experience than GPs in dealing with these issues. [This helps] especially when you are kind of insecure as a GP." (R17)*

However, all participants argued that GPs have a distinct part to play in suicide prevention, because, if GPs do not recognise patients' psychological complaints, patients may not get a referral at all to neither MHSS nor MHC services.

## Implementation process

SUPRANET for PCPs consists of a face-to-face training and some additional components. Of all the PCPs who participated in this study, very few had experienced all the components; only two were involved in improving continuity of care and only some used the additional materials. All PCPs participated in the suicide prevention training.

**Engaging GP practices in SUPRANET.**   Regions were responsible for contacting and inviting GP practices to participate, which was a challenging task because of PCPs' limited

available time. One PCP questioned whether the project attracted the right kind of participants using this strategy.

> "Only PCPs who are interested in the topic, will sign up for such a program. They often have a background in or affinity with the topic, and, therefore, perform better when faced with these problems during a consultation. The ones who need it the most, who don't have good communication skills or an affinity with the problem, will not sign up voluntarily". (R19)

Therefore, this PCP suggested offering the training during the contracted training hours of GP practice staff, so that it is harder for PCPs to avoid the training and it might facilitate implementation.

One of the non-clinical SUPRANET professionals argued they could have used more support during the recruitment phase. When this region involved MHC services to engage GP practices things started to move. Various reasons were offered for this effect: PCPs may favour being approached by other medical doctors, MHC services offered the training for free, and the training was reduced to one or two hours instead of four. An advantage of this approach was instant contact between primary care and MHC, which also facilitated communication and collaboration in daily practice. However, this approach also had some important disadvantages, since there was less control and continuity with regard to the content of the training.

**Leadership of SUPRANET.**   In most regions, a PCP, often a GP, functioned as implementer and spokesperson for SUPRANET. This PCP was responsible for bringing together colleagues from various practices and logistically organising the suicide prevention training. In addition, some of these PCPs were also involved in the continuity of care aspect of SUPRANET. This latter demanded a lot of time and energy and these PCPs argued they expected more support and cooperation from both MHC services and 113 Suicide Prevention. In the absence of this, one PCP felt isolated and became demotivated.

> "I've sort of lost my motivation. [. . .] I don't see it happening anymore. I have really put a lot of time into it. [. . .] It feels like it was my request to start this project, but that is not the case. We have been asked by 113 Suicide Prevention, [. . .] and we said yes, but on the assumption that MHC services were also taking part". (R2)

**SUPRANET as strategy for applying suicide prevention practices.**   Almost all PCPs evaluated the suicide prevention training positively. Although they stated it did not offer any ground-breaking insights, the majority thought it had increased their knowledge about and awareness of suicide prevention. In addition, some argued that SUPRANET had made them feel better equipped to assess patients' suicide risk:

> "Since the training, I ask more often and more targeted questions about people's emotions and their suicide risk, sometimes very directly, like, 'Would you take your life?' [. . .] I did that before as well, but because of the training you become aware of certain aspects of your practice you can improve." (R1)

Further, the PCPs thought it was valuable to discuss clinical cases with colleagues, since there is usually no time for this during practice hours. They also appreciated being able to practice communication skills during the training and learned, for example, the importance of making actual contact with patients:

*"Asking about the patient's feelings of despair and just carefully listen, without thinking of a solution straight away, were two eye-openers, which I applied the next day. [. . .] and it worked. The lady said: 'I am so happy that I could tell my story, that you just listened to me'. So, I thought, 'This works well'." (R18)*

However, there was also some critique on the training, with regard to both content and form. Some PCPs wished the training included more new scientific knowledge and insights. One PCP reported that he would have liked to learn more about risk groups and when to discuss suicidal feelings with patients, instead of how. Other suggestions included incorporating more information about discussing suicidal feelings with specific subgroups, such as children and adolescents, patients with different cultural backgrounds or bereaved family members.

Initially, PCPs thought that conducting four hours of training outside office hours would be too demanding. Afterwards, most were content with it and considered it as something you should prioritize.

*"I get it, people want it to be as quick and efficient as possible, but, in terms of effectiveness I think this [the duration] was good. I understand that people want it shorter; everybody is busy, they always are. But I think you should make time for it. There are certain things you just have to make time for." (R19)*

However, since PCPs initially felt resistance towards the duration, they suggested to shorten the training or to split it up into two parts: an individual online training complemented with a face-to-face group training to practice communication skills. The PCPs also strongly recommended follow-up training, since they believed it is not only important to raise awareness but also to maintain it.

The additional materials (checklist, information material, flyers and posters and information module on suicide in relation to medication) were rated by some participants as helpful. Some argued, for instance, the checklist served as a reminder for asking all the relevant questions. Others thought the information material was useful for disseminating knowledge to colleagues. The poster and flyers were, to the PCPs' knowledge, never used by patients as an encouragement to discuss suicidal feelings. Although not many PCPs completed the module on medication and suicide because of the strict scheduling of training time within their GP practice, they thought this module would be very relevant because not all PCPs have the same knowledge with regard to medication. The continuity of care was assessed as an important component of SUPRANET, which required a lot of improvement, though hardly any PCPs experienced this due to lagging implementation.

## Discussion

In this qualitative study, we evaluated part of a multilevel suicide prevention program (SUPRANET), which was implemented in various regions in the Netherlands, as strategy for applying suicide prevention practices in primary care. The CFIR was used to structure the data and to gain insights in facilitating and limiting factors of SUPRANET. All but one domain (individual characteristics) from the CFIR were explicitly addressed. Since this study was based on PCPs' perceptions and experiences, individual characteristics are intertwined with other domains, particular with intervention characteristics. This study found that there were important barriers to the assessment of suicide risk in patients (intervention characteristics) because of PCPs' self-perceived incompetence, burdensomeness of suicide and their lack of time and workload. The relationship with MHC was an important limiting factor from the

outer setting and with regard to inner setting, MHSS had a positive influence on implementation. Although there were some difficulties in the process of implementation, SUPRANET itself was perceived as useful strategy to improve suicide prevention practices.

Persistent challenges for suicide prevention include the lack of competence to assess suicide risk. In the literature, PCPs also reported challenges with regard to exploring suicidal feelings, because of the perceived reluctance of their patients to disclose suicidal ideation [30] and because many believed that suicide is difficult to predict and prevent [31, 32]. Lack of time is an universally acknowledged barrier to suicide prevention [31]. The present study found that time and workload were limitations for both participating in SUPRANET and for assessing and managing suicide risk with patients.

Hegerl et al. [9] also described in their overall evaluation of the EAAD model, which is the foundation of SUPRANET, that it required significant effort to train GPs. Eventually, 304 GPs from four countries participated in the training, but only after having made considerable adjustments, such as shortening or obliging the training. Another study reported significant improvement in GPs' attitudes and confidence straight after this training, but only improved confidence was maintained after three to six months [33]. In Hungary, GPs' need for referral options was not addressed, which, according to the authors, may explain the lack of sustainable changes [9]. Within SUPRANET, the training had been delivered to 67 PCPs by the time of the study's deadline (March 2018), but delivery continued within these and other regions. GPs were often reluctant to participate because of a lack of time and a heavy workload. However, as PCPs from this study mentioned, participating is not just a matter of time but also a matter of priority.

In the literature, providing support to PCPs is described as one of the most important suicide prevention strategies [5, 16, 17]. On the other hand, Milner et al. [34] argued that many of the included studies were biased and their results varied by study design and outcome. They recommended more studies with randomized, controlled, and blinded designs, as well as establishing effects on various outcome variables, before widely rolling out GP suicide prevention initiatives. Although this study did not meet these requirements, it showed that PCPs valued the training and that it positively influenced their attitudes, suicide assessment and clinical management skills, effects that have also been reported by previous studies [33, 35–37]. Nevertheless, the present study also shows that exploration of suicidal feelings remains difficult and that providing education alone is not sufficient for effective suicide prevention. This is confirmed by Gask who, with regard to educating GPs to recognize and manage depression, states that providing education alone, although necessary, is not sufficient. It is important to make sure that both the content and the provision method of education material suits people at different stages of readiness [38].

The PCPs in the present study expressed a pressing need to improve access to and collaboration with MHC services. The lack of such collaboration is a common barrier to suicide prevention in countries with a primary health care system. In the United Kingdom, for example, many studies have described the issues that GPs have with MHC services. Among these are the tendency of MHC professionals to minimize GPs' assessments of patients' suicidal state. GPs report feeling stuck with patients, because they rarely meet the criteria for review and, therefore, remain in primary care [31, 39]. A recent British study described GPs' feelings of professional isolation as being "lost in a referral maze" [40 p. 5]. British GPs have also expressed the need to have mental health staff based in GP practices [40], such as the MHSS in the Netherlands' health care system. In the present study, this was indeed perceived as an immense improvement to the management of suicidal patients and it was suggested to expand their role with regard to suicide prevention.

## Limitations

The study had several limitations. Since participation in SUPRANET was voluntary, it is likely SUPRANET attracted especially PCPs with a special interest in suicide prevention. Most of the PCPs of this study indeed reported a special interest or experience in suicide prevention or mental health issues. This inherent bias in the sampling method means that the outcomes of the study may not be generalizable to the entire PCP population. Other factors, difficulties, or needs might be identified among PCPs who do not have an interest or experience in the topic. However, it is a well-known phenomenon in implementation studies that these are often based on early adopters. This does not necessarily mean that the issues that are reported here are not applicable to other PCPs. Another limitation is the number of PCPs participating in this study. We could only invite PCPs who participated in SUPRANET and since not all wanted to be interviewed, we eventually included 18 PCPs. In qualitative designs, it is quite normal to have a small sample size, since the aim is to provide rich insights rather than statistical information. There is no rule for determining the sample size of qualitative studies, but similar issues and themes were identified throughout the interviews, indicating that it was sufficient [41].

## Implications for practice

The results from the interviews have been discussed with the implementation team of the 113 Suicide prevention, which will continue to implement SUPRANET in new regions in the Netherlands. Regions will now receive more guidance during the recruitment phase, and they will be encouraged to increase the role of MHSS since that may improve the implementation process and the collaboration with MHC. Besides, regions will be urged to involve MHC services from the beginning. Once started, specific focus will be devoted to the continuity of care element, for which a new format has been created: '*work session*: *collaborating and making agreements with MHC*'. This was developed by the Dutch College of General Practitioners and consists of meetings with relevant stakeholders within the regional chain of care (GPs, MHSS, MHC services and possibly district teams). These meetings will be chaired by an independent party, who will steer towards mutual agreements about the treatment and management of suicidal patients [42]. Some of the proposed adjustments to the training have been carried out already, such as the development of a two-part training. An e-learning program was developed and launched in October 2019 and is freely available for all PCPs. The e-learning can be complemented with group-based skills training to explore suicidal feelings and discuss clinical cases with colleagues, which is strongly recommended. Given the potential of PCPs and their experienced barriers for effective suicide prevention, we encourage that national governments and professional PCP associations devote attention to suicide prevention in primary care, help addressing these challenges and put efforts into engaging PCPs more effectively.

## Conclusion

This study is among the few that describes experiences and perceptions with regard to a suicide prevention program for primary care as part of a multilevel community-based model for suicide prevention. The study provides deeper understanding of how PCPs can be better engaged in suicide prevention. SUPRANET was perceived as useful strategy to improve suicide prevention practices. The training increased PCPs' awareness and knowledge of suicide prevention, which facilitates the recognition of suicidal patients. Effective suicide prevention, however, also requires improved collaboration between primary care and MHC. More effectively addressing this will facilitate implementation and effectiveness of SUPRANET and further engage PCPs in suicide prevention.

## Supporting information

**S1 Material. Topic list.**
(DOCX)

## Acknowledgments

We thank all the participants for their time and input.

## Author Contributions

**Conceptualization:** Elke Elzinga, Anja J. T. C. M. de Kruif, Derek P. de Beurs, Aartjan T. F. Beekman, Renske Gilissen.

**Formal analysis:** Elke Elzinga, Anja J. T. C. M. de Kruif.

**Funding acquisition:** Derek P. de Beurs, Gerdien Franx, Renske Gilissen.

**Methodology:** Elke Elzinga, Anja J. T. C. M. de Kruif, Derek P. de Beurs, Aartjan T. F. Beekman, Renske Gilissen.

**Project administration:** Elke Elzinga.

**Supervision:** Anja J. T. C. M. de Kruif, Derek P. de Beurs, Aartjan T. F. Beekman, Renske Gilissen.

**Validation:** Derek P. de Beurs, Gerdien Franx, Renske Gilissen.

**Writing – original draft:** Elke Elzinga, Anja J. T. C. M. de Kruif, Derek P. de Beurs, Aartjan T. F. Beekman, Gerdien Franx, Renske Gilissen.

**Writing – review & editing:** Elke Elzinga, Anja J. T. C. M. de Kruif, Derek P. de Beurs, Aartjan T. F. Beekman, Gerdien Franx, Renske Gilissen.

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
