## [Decision Letter · Decision Letter 0]

3 Jun 2020

PONE-D-20-01479

Engaging Primary Care Professionals in suicide prevention: a qualitative study

PLOS ONE

Dear Dr. Elzinga,

Thank you for submitting your manuscript to PLOS ONE. I have recently taken over this manuscript as Academic Editor and have had the opportunity to review the comments provided by the expert reviewer that the prior Academic Editor invited. I have also had a chance to review your paper as well. As you will see below, the reviewer had a generally positive opinion of the paper but had concerns that are likely addressable. Accordingly, I am inviting you to revise and resubmit your manuscript. Please do know that this is no guarantee of ultimate acceptance and I will send the revised version of this manuscript back to the original reviewer for comments again. 

Although all of the reviewer's comments are important to address, there are a few comments in particular that I'd like you to pay attention to. Specifically, it would be good to add more detail on (a) the background of how PCPs are trained in suicide screening/prevention/intervention and (b) what the specific suicide screening/prevention/intervention programs that are implemented are. Additionally, as the reviewer notes, there are existing research frameworks where your study fits well (e.g,. The Consolidated Framework for Implementation Research) and it would be good to discuss this in your introduction. 

We look forward to receiving your revised manuscript.

Kind regards,

Evan M Kleiman

Academic Editor

PLOS ONE

Journal Requirements:

2. For qualitative studies, PLOS ONE suggests consulting the COREQ guidelines: http://intqhc.oxfordjournals.org/content/19/6/349 to ensure that all relevant information is provided (in this case we would appreciate more information about: the number  and training of interviewers; how participants were selected; if bias issues were considered.

3. Please include the registration number for the clinical trial referenced in the manuscript.

Reviewers' comments:

Reviewer's Responses to Questions

Comments to the Author

1. Is the manuscript technically sound, and do the data support the conclusions?

Reviewer #1: Partly

2. Has the statistical analysis been performed appropriately and rigorously? 

Reviewer #1: Yes

3. Have the authors made all data underlying the findings in their manuscript fully available?

Reviewer #1: No

4. Is the manuscript presented in an intelligible fashion and written in standard English?

Reviewer #1: No

5. Review Comments to the Author

Reviewer #1: Thank you for the opportunity to review this manuscript. This paper presents the results of qualitative interviews with 21 individuals involved in the implementation of a primary care based suicide prevention program, a timely and important topic. I have several concerns about the manuscript in its current form, detailed below, that I think would be helpful for the authors to address.

While the manuscript is generally well written, additional proofreading by a native English speaker would be helpful.

Perhaps it is just a problem with the proof, but the text in Box 1 appears cut off.

It might be helpful to describe the suicide screening/prevention/intervention practices being implemented as the “interventions” and the additional training and support being offered to the clinicians as the “implementation strategies.” It would also be helpful to understand the intensity of training and supports provided to PCPs. Powell et al’s (2015) refined compilation of implementation strategies paper and Proctor et al.’s (2013) paper on specifying and reporting implementation strategies, both published in Implementation Science, may be helpful in these regards. I think this approach would improve clarity throughout.

The authors are encouraged to consider making the “topic list” used to guide interviews available as supplemental to the paper. What was the rationale for this approach versus a more traditional semi-structured qualitative interview guide? A semi-structured guide has several advantages including ensuring uniform inclusion and sequencing of questions.

If there is any additional demographic information about participants please consider including it, for example participant race, ethnicity, mean age, mean number of years in practice, and information about the types of practices (community clinics vs. academically affiliated centers for example).

Some of the content of the results seems to report perceptions or suggestions made by one PCP. While it is certainly appropriate to provide illustrative quotations from participants, the reader can be much more confident in the trustworthiness of overall themes that emerged based on the perspectives of many of the interviewees and not individual interviewee perspectives. It may be that this is simply an issue with how the results were presented in the paper and not the actual coding process.

What is 113 referring to in the Results? Is this masking a person or program or an actual name of a program?

Consider grounding this work in an established implementation framework, like the Consolidated Framework for Implementation Research (CFIR). Most of the themes fit within established constructs from frameworks like CFIR and organizing and presenting them in this way could be useful. An integrated coding approach (see Bradley) that combines inductive and deductive coding could offer advantages to solely utilizing grounded theory here. It could be helpful to review the CFIR implementation process constructs in particular to see if those may be useful for organizing the presentation of the results around providers experience with implementation. Another suggestion would be to specifically pull out barriers, facilitators, and suggested strategies/changes and present as their own themes. Presently these are presented within other themes.

Finally, I see that e-learning is mentioned in the discussion as a strategy that was suggested by providers, but I don’t recall seeing that presented in the results. I suggest ensuring that all themes commented on in the discussion are reported in the results.

6. PLOS authors have the option to publish the peer review history of their article (what does this mean?). If published, this will include your full peer review and any attached files.

Do you want your identity to be public for this peer review? For information about this choice, including consent withdrawal, please see our Privacy Policy.

Reviewer #1: No

---

## [Author Response · Author response to Decision Letter 0]

17 Jul 2020

Dear Dr. Elzinga,

Thank you for submitting your manuscript to PLOS ONE. I have recently taken over this manuscript as Academic Editor and have had the opportunity to review the comments provided by the expert reviewer that the prior Academic Editor invited. I have also had a chance to review your paper as well. As you will see below, the reviewer had a generally positive opinion of the paper but had concerns that are likely addressable. Accordingly, I am inviting you to revise and resubmit your manuscript. Please do know that this is no guarantee of ultimate acceptance and I will send the revised version of this manuscript back to the original reviewer for comments again. 

Although all of the reviewer's comments are important to address, there are a few comments in particular that I'd like you to pay attention to. Specifically, it would be good to add more detail on (a) the background of how PCPs are trained in suicide screening/prevention/intervention and (b) what the specific suicide screening/prevention/intervention programs that are implemented are. Additionally, as the reviewer notes, there are existing research frameworks where your study fits well (e.g,. The Consolidated Framework for Implementation Research) and it would be good to discuss this in your introduction. 

If applicable, we recommend that you deposit your laboratory protocols in protocols io to enhance the reproducibility of your results. Protocols.io assigns your protocol its own identifier (DOI) so that it can be cited independently in the future. For instructions see: http://journals.plos.org/plosone/s/submission-guidelines#loc-laboratory-protocols

We look forward to receiving your revised manuscript.

Kind regards,

Evan M Kleiman

Academic Editor

PLOS ONE

Dear dr. Evan Kleiman,

Thank you for providing the opportunity to revise the manuscript and considering it for publication in PLOS ONE. We believe that the revisions have improved the manuscript considerably. We have addressed your comments below using italic script.

Journal Requirements:

Thank you, I have adjusted the manuscript according to the style requirements.

2. For qualitative studies, PLOS ONE suggests consulting the COREQ guidelines: http://intqhc.oxfordjournals.org/content/19/6/349 to ensure that all relevant information is provided (in this case we would appreciate more information about: the number and training of interviewers; how participants were selected; if bias issues were considered.

Thank you for this comment. We have carefully read the COREQ guidelines and added information about the number and training of interviewers, participant selection and if biases were considered.

Line 126-128: “The interviews were conducted by one researcher (EE), who had completed certified training in qualitative research methods including interview skills, in the period October 2018 to March 2019.”

Line 105-110: “We used convenience sampling and the snowball technique to recruit study participants from the PCPs within the SUPRANET regions who completed the suicide prevention training. We reached out to PCPs directly or to contact persons in GP practices via email and by phone. In regions where implementation was lagging, i.e., where no PCPs were being trained before the study deadline, professionals who were involved with the implementation of the SUPRANET intervention were approached for an interview.”

Line 471-476: “Since participation in the intervention and evaluation study was voluntary, it is likely that especially PCPs with a special interest in suicide prevention predominated among those who signed up. Most of the PCPs of this study indeed reported they had a special interest or experience in suicidality or mental health issues, for example, by having worked in a MHC institution. This inherent bias in the sampling method means that the outcomes of the study may not be generalizable to the entire PCP population.”

3. Please include the registration number for the clinical trial referenced in the manuscript.

We have not referred to a clinical trial in this manuscript.

The paper includes anonymized quotes of the interviewees to support the results. It is not possible to make the full transcripts of the interviews publicly available. The interviewees have not given consent for this, and they may contain sensitive or identifiable information. The transcripts are archived on a secured server of 113 Suicide Prevention. Anonymized excerpts of transcripts are available upon request, which can be made via email to the corresponding author (Elke Elzinga; e.elzinga@113.nl) or to the research department of 113 Suicide Prevention (onderzoek@113.nl). 

We have included captions for all tables and excluded the figure from the manuscript.

 

Reviewers' comments:

Reviewer's Responses to Questions

Comments to the Author

1. Is the manuscript technically sound, and do the data support the conclusions?

Reviewer #1: Partly

 2. Has the statistical analysis been performed appropriately and rigorously? 

Reviewer #1: Yes

3. Have the authors made all data underlying the findings in their manuscript fully available?

Reviewer #1: No

4. Is the manuscript presented in an intelligible fashion and written in standard English?

Reviewer #1: No

5. Review Comments to the Author

Reviewer #1: Thank you for the opportunity to review this manuscript. This paper presents the results of qualitative interviews with 21 individuals involved in the implementation of a primary care based suicide prevention program, a timely and important topic. I have several concerns about the manuscript in its current form, detailed below, that I think would be helpful for the authors to address.

Dear reviewer,

Thank you for you carefully formulated comments. We have addressed them in the manuscript and in this letter, using italic script.

While the manuscript is generally well written, additional proofreading by a native English speaker would be helpful.

Thank you for your suggestion, the manuscript is now proofread by a professional language bureau.

Perhaps it is just a problem with the proof, but the text in Box 1 appears cut off.

Thank you for noticing, the box was indeed cut off. We have deleted the box and incorporated it in the introduction.

Line 65-69: “Since 2007, GPs can also refer patients to the Mental Health Nurse (MHN), a role introduced to cope with the increasing demand for mental health issues in primary care [21]. MHNs include professionals with a nursing or psychology background who can provide brief therapy sessions to patients within primary care [10].” 

It might be helpful to describe the suicide screening/prevention/intervention practices being implemented as the “interventions” and the additional training and support being offered to the clinicians as the “implementation strategies.” It would also be helpful to understand the intensity of training and supports provided to PCPs. Powell et al’s (2015) refined compilation of implementation strategies paper and Proctor et al.’s (2013) paper on specifying and reporting implementation strategies, both published in Implementation Science, may be helpful in these regards. I think this approach would improve clarity throughout.

Thank you for this suggestion. We have clarified the terms by referring to “intervention” with regard to the entire package, including the suicide prevention training, continuity of care element, module on suicide and medication, suicide prevention guide, flyers and posters and checklist (see table 1 below), and by referring to “implementation strategies” with regard to offering the intervention to the clinicians all over the manuscript. We have also adjusted the table and added extra information about what these elements entail and described the intensity of the elements. 

Line 102:

Table 1. Various elements of the SUPRANET intervention for PCPs

Suicide prevention training 

The training consists of a theoretical section covering the epidemiology, behaviour, and process of suicide, the Chronological Assessment of Suicidal Events (CASE) methodology, and treatment and referral. The tuition is interspersed with interactive exercises focused on making contact with feelings of despair, involving relatives, and diagnosis. The training is provided by experienced trainers from the Dutch College of General Practitioners and is accredited for four hours.

Continuity of care 

CPs are encouraged to arrange meetings with organizations in the regional chain of care for suicidal patients (e.g., MHC institutions, crisis services, or emergency rooms) to discuss issues and to enter into agreements to improve the continuity of care.

Module on suicide and medication 

A two-hour (accredited) individual e-learning session or a pharmaceutical therapeutic audit completed with colleagues takes place on the subject of the role of medication in suicide prevention.

Suicide prevention guide 

The suicide prevention guide contains a summary of the most important information from the suicide prevention training and the multidisciplinary guideline for the diagnosis and treatment of suicidal behaviour including a triage tool to help in referrals of suicidal patients.

Flyers and posters 

Flyers and posters are aimed at patients and relatives, to encourage them to talk about and seek help for suicidal feelings.

Checklist 

The checklist can be completed after consultations with (possible) suicidal patients. The checklist includes items such as assessment of suicidal feelings, rumination, and the concreteness of suicidal plans.

The authors are encouraged to consider making the “topic list” used to guide interviews available as supplemental to the paper. What was the rationale for this approach versus a more traditional semi-structured qualitative interview guide? A semi-structured guide has several advantages including ensuring uniform inclusion and sequencing of questions.

Thank you for this suggestion. We have added the topic list in the supplementary files (S1 File. Topic list) and explained the reasons for this approach in the method section. 

Line 114-119: “A topic list was used to guide the semi-structured interviews (See S1. Topic list). This instrument consists of a list of topics and prompts based on literature [24–26] and discussed within the research group (AK, DB, RG, EE). The rationale for this approach is that it is best suited to capture the perceptions and experiences of participants [27]. Although we mostly addressed the topics in the given order, if required, we could discuss them in another order, and it leaves space for participants to talk freely and bring up topics or issues themselves”

If there is any additional demographic information about participants please consider including it, for example participant race, ethnicity, mean age, mean number of years in practice, and information about the types of practices (community clinics vs. academically affiliated centers for example). 

Unfortunately, there is not. We only have information regarding function and gender (and region, but that cannot be published due to privacy restrictions).

Some of the content of the results seems to report perceptions or suggestions made by one PCP. While it is certainly appropriate to provide illustrative quotations from participants, the reader can be much more confident in the trustworthiness of overall themes that emerged based on the perspectives of many of the interviewees and not individual interviewee perspectives. It may be that this is simply an issue with how the results were presented in the paper and not the actual coding process.

Thank you for your suggestion. The results are not based upon the data from one PCP, but on all 21 participants. We have included numbers of the interviewees to the quotes, hopefully this clarifies that the comments originated from different participants.

What is 113 referring to in the Results? Is this masking a person or program or an actual name of a program?

113 refers to 113 Suicide prevention: the initiator of SUPRANET. We have now addressed this (more clearly) in the method section. 

Line 85-86: “SUPRANET was initiated by 113 Suicide Prevention, the Netherlands national suicide prevention centre, as part of the national agenda for suicide prevention commissioned by the Ministry of Health”. 

Consider grounding this work in an established implementation framework, like the Consolidated Framework for Implementation Research (CFIR). Most of the themes fit within established constructs from frameworks like CFIR and organizing and presenting them in this way could be useful. 

We have carefully read the paper about the CFIR and it seems to provide a very relevant framework for when performing an implementation study. In this evaluation study, where we focused on the perceptions and experiences of PCPs with the intervention, we did not select a framework to guide the data collection and analysis procedure upfront. Therefore, we chose to discuss the CFIR it in the discussion.

Lines 407-426: “Although we did not select a framework a priori, using a framework, such as Damschroder and colleagues’ Consolidated Framework for Implementation Research (CFIR), may aid in the evaluation of complex interventions such as SUPRANET [34]. The CFIR consists of five domains (intervention characteristics, outer setting, inner setting, characteristics of individuals, and process) that interact in various ways and determine the effectiveness of implementation. When applying the CFIR retrospectively, we were able to establish some factors that may have played a crucial role in the effectiveness of the intervention. The suicide prevention training can be seen as the “core component” with an “adaptable periphery” consisting of the other elements and its flexible delivery. According to the CFIR, this facilitates implementation and thus influences effectiveness. Further, peer pressure from other PCPs, perceived patients’ needs, and time constraints can be identified as factors from the outer setting that influence participation. Factors from the inner setting include a positive climate for implementation within the GP practices, shown for example by PCPs who wanted to participate because of experiences with a patient’s suicide (tension for change), because they wanted to develop their skills (positive learning climate), or because of the perceived urgency of the topic (relative priority). This study also indicated that some individuals, notably key PCPs, played an important role in the implementation of the intervention. The same applies in reverse; implementation in certain regions may have lagged behind others because no such individuals were driving the initiative. Although we could apply this framework retrospectively, more or other factors may have come up when applying it from the beginning. Therefore, future evaluation studies are recommended to use this or another framework to guide the evaluation from the start.”

• An integrated coding approach (see Bradley) that combines inductive and deductive coding could offer advantages to solely utilizing grounded theory here. 

In their article Bradley et al 2007 focus on qualitative data analysis that has as a goal the generation of taxonomy, themes, and theory germane to health services research. Because of the explorative nature of our research we conducted a grounded theory approach. Our approach was inductive and data-driven, focused upon identifying and discussing the salient themes repeated across and within transcripts. It involves coding and the generation (and interpretation) of broader patterns in data. Our choice is therefore prompted by carrying out the analysis from this point of view.

• It could be helpful to review the CFIR implementation process constructs in particular to see if those may be useful for organizing the presentation of the results around providers experience with implementation. 

Thank you for your suggestion. Even though evaluation the implementation process was not the main goal of the paper, implementation-related themes did emerge from the data which we addressed in the chapter ‘implementation strategies’ (lines 165-207). Further, we discussed the application of the CFIR framework onto our results in the discussion. From the implementation construct, we were especially able to establish the role of certain individuals (key PCPs) during the implementation process.

Line 421-425: “This study also indicated that some individuals, notably key PCPs, played an important role in the implementation of the intervention. The same applies in reverse; implementation in certain regions may have lagged behind others because no such individuals were driving the initiative. Although we could apply this framework retrospectively, more or other factors may have come up when applying it from the beginning.” 

• Another suggestion would be to specifically pull out barriers, facilitators, and suggested strategies/changes and present as their own themes. Presently these are presented within other themes.

Thank you for this suggestion. However, the rationale of this study was to provide insights into the perceptions and experiences of PCPs with the intervention and suicide prevention in general practice, which will ultimately be used to enhance the intervention and to engage PCPs more effectively in suicide prevention. The topic list that we used in this study was based on this rationale. Although barriers and facilitators emerged, we did not set up this study to focus on those specifically. It should be noted that the barriers and facilitators that are being described in the present study emerged within the context of the themes and are non-exhaustive. From this perspective, we think the chosen structure into the main themes is most suitable.

Finally, I see that e-learning is mentioned in the discussion as a strategy that was suggested by providers, but I don’t recall seeing that presented in the results. I suggest ensuring that all themes commented on in the discussion are reported in the results.

Thank you, we have described this in the result section:

Line 238-239: “Other suggestions included shortening the training or splitting it up into two parts: online individual training complemented with training in a group to practice communication skills.

6. PLOS authors have the option to publish the peer review history of their article (what does this mean?). If published, this will include your full peer review and any attached files.

Do you want your identity to be public for this peer review? For information about this choice, including consent withdrawal, please see our Privacy Policy.

Reviewer #1: No

Once again, thank you for your comments. We believe that with these revisions the manuscript has improved a lot.

---

## [Decision Letter · Decision Letter 1]

9 Sep 2020

PONE-D-20-01479R1

Engaging Primary Care Professionals in suicide prevention: a qualitative study

PLOS ONE

Dear Dr. Elzinga,

Thank you for submitting your manuscript to PLOS ONE. After careful consideration, we feel that it has merit but does not fully meet PLOS ONE’s publication criteria as it currently stands. Therefore, we invite you to submit a revised version of the manuscript that addresses the points raised during the review process.

As you will see below, the reviewer still has several lingering concerns. I share these concerns as well and am, at this point, uncertain about whether or not it will be possible to revise the paper to correct these issues. I would like to give you another opportunity to address the issues and respond to the reviewer's very detailed feedback. If you are able to do this, there is a chance that the paper may be fit for PLOS One. But, I should say that I will completely understand if you choose to instead submit this paper to another outlet. If you do choose to submit elsewhere, please let me know. 

If you do choose to resubmit, please submit your revised manuscript by Oct 24 2020 11:59PM. If you will need more time than this to complete your revisions, please reply to this message or contact the journal office at plosone@plos.org. Please include the following items when submitting your revised manuscript:

We look forward to receiving your revised manuscript.

Kind regards,

Evan M Kleiman

Academic Editor

PLOS ONE

Reviewers' comments:

Reviewer's Responses to Questions

**Comments to the Author**

1. If the authors have adequately addressed your comments raised in a previous round of review and you feel that this manuscript is now acceptable for publication, you may indicate that here to bypass the “Comments to the Author” section, enter your conflict of interest statement in the “Confidential to Editor” section, and submit your "Accept" recommendation.

Reviewer #1: (No Response)

2. Is the manuscript technically sound, and do the data support the conclusions?

Reviewer #1: Yes

3. Has the statistical analysis been performed appropriately and rigorously? 

Reviewer #1: Yes

4. Have the authors made all data underlying the findings in their manuscript fully available?

Reviewer #1: Yes

5. Is the manuscript presented in an intelligible fashion and written in standard English?

Reviewer #1: Yes

6. Review Comments to the Author

Reviewer #1: Thank you for the opportunity to re-review this manuscript. I appreciate the detailed rationale provided in the authors’ response to reviews. There are a few remaining issues that warrant consideration.

Minor point- Table 2. I do not think this level of detail about participants is necessary, rather I would suggest reporting in aggregate the n (%) of participants who are male, GPs etc.

There remains some confusion with respect to the intervention vs. implementation strategies. In the way these terms are typically used, the interventions are the evidence based suicide prevention strategies that the PCPs use with their patients. The implementation strategies are the strategies employed in order to get PCPs to implement EBPs for suicide prevention, such as training, consultation, facilitation etc.

Geoff Curran has a paper on the topic that very clearly describes the distinction: Curran, G.M. Implementation science made too simple: a teaching tool. Implement Sci Commun 1, 27 (2020). https://doi.org/10.1186/s43058-020-00001-z

Specifically, it would be helpful to clearly distinguish participants’ reactions to the SUPRANET training/supports (the implementation strategies) vs. their reactions to implementing EBPs for suicide prevention (the intervention).

I think the way the CFIR is integrated at present is confusing. I understand you do not wish to retrospectively apply CFIR to your data and that is fine, however if that is the case then I would not attempt to do that in the Discussion as you did. Instead, I would suggest removing CFIR from the Discussion and incorporating it into the Introduction. I do believe that this is in fact an implementation study- you are studying an effort to train providers to implement suicide prevention practices, exploring PCPs reactions and utilization of suicide prevention practices, etc. In contrast, an intervention study would typically involve studying the impact of suicide prevention practices on patients. Again, Geoff Curran’s paper may be helpful clarification.

Given this, CFIR seems quite relevant. In the Introduction I would suggest you introduce the CFIR as a comprehensive framework that describes the relevant contextual factors that have been shown to be important to consider when planning for the implementation of a health service intervention (e.g., suicide prevention EBPs). Then, I would briefly describe the 5 main CFIR constructs and delineate which you focused on in the present project. For example, Characteristics of the Intervention (e.g., the characteristics of suicide prevention EBPs that may impact their use in a particular setting or by particular providers), Outer Setting (e.g., external factors such as political or policy factors related to suicide prevention, patient populations served by the PCPs), Inner Setting (e.g., what is it about the way clinics are structured or ways in which workflows or availability of behavioral health consultant impacts practice?), Characteristics of the Individuals implementing suicide EBPs (e.g., PCP attitudes about suicide prevention, self-efficacy to implement suicide EBPs) and Implementation Process (e.g., what was the experience of receiving training through SUPRANET like).

I hope this helps to clarify some of my previous comments from the initial submission.

7. PLOS authors have the option to publish the peer review history of their article (what does this mean?). If published, this will include your full peer review and any attached files.

Reviewer #1: No

---

## [Author Response · Author response to Decision Letter 1]

23 Oct 2020

Dear editor and reviewer, 

Thank you for the opportunity to revise the manuscript once more. We have discussed your comments thoroughly within our research team and addressed these in the manuscript. Below, we will provide an answer to the issues one-by-one:

Reviewer #1: Thank you for the opportunity to re-review this manuscript. I appreciate the detailed rationale provided in the authors’ response to reviews. There are a few remaining issues that warrant consideration.

Minor point- Table 2. I do not think this level of detail about participants is necessary, rather I would suggest reporting in aggregate the n (%) of participants who are male, GPs etc.

Thank you for your comment, we have adjusted the table (line 149).

There remains some confusion with respect to the intervention vs. implementation strategies. In the way these terms are typically used, the interventions are the evidence based suicide prevention strategies that the PCPs use with their patients. The implementation strategies are the strategies employed in order to get PCPs to implement EBPs for suicide prevention, such as training, consultation, facilitation etc.

Geoff Curran has a paper on the topic that very clearly describes the distinction: Curran, G.M. Implementation science made too simple: a teaching tool. Implement Sci Commun 1, 27 (2020). https://doi.org/10.1186/s43058-020-00001-z

Specifically, it would be helpful to clearly distinguish participants’ reactions to the SUPRANET training/supports (the implementation strategies) vs. their reactions to implementing EBPs (evidence-based practices) for suicide prevention (the intervention).

Thanks for clearing up the confusion. We have changed this throughout the manuscript; SUPRANET is now described as implementation strategy to help PCPs applying suicide prevention practices. See for example the altered aim on line 75-78:

“Using a qualitative design, we conducted this implementation study to evaluate to what extent SUPRANET was helpful in supporting PCPs to apply suicide prevention practices. These insights will be used to engage PCPs more effectively in suicide prevention by improving the use of SUPRANET.”

I think the way the CFIR is integrated at present is confusing. I understand you do not wish to retrospectively apply CFIR to your data and that is fine, however if that is the case then I would not attempt to do that in the Discussion as you did. Instead, I would suggest removing CFIR from the Discussion and incorporating it into the Introduction. I do believe that this is in fact an implementation study- you are studying an effort to train providers to implement suicide prevention practices, exploring PCPs reactions and utilization of suicide prevention practices, etc. In contrast, an intervention study would typically involve studying the impact of suicide prevention practices on patients. Again, Geoff Curran’s paper may be helpful clarification.

Given this, CFIR seems quite relevant. In the Introduction I would suggest you introduce the CFIR as a comprehensive framework that describes the relevant contextual factors that have been shown to be important to consider when planning for the implementation of a health service intervention (e.g., suicide prevention EBPs). Then, I would briefly describe the 5 main CFIR constructs and delineate which you focused on in the present project. For example, Characteristics of the Intervention (e.g., the characteristics of suicide prevention EBPs that may impact their use in a particular setting or by particular providers), Outer Setting (e.g., external factors such as political or policy factors related to suicide prevention, patient populations served by the PCPs), Inner Setting (e.g., what is it about the way clinics are structured or ways in which workflows or availability of behavioral health consultant impacts practice?), Characteristics of the Individuals implementing suicide EBPs (e.g., PCP attitudes about suicide prevention, self-efficacy to implement suicide EBPs) and Implementation Process (e.g., what was the experience of receiving training through SUPRANET like).

I hope this helps to clarify some of my previous comments from the initial submission.

Thanks for your explanation. We have now introduced the CFIR in the introduction, explained how we apply the model in the method-section and we have organized the results according to the CFIR structure. In the discussion we present the most important findings according to the CFIR (see below).

Introduction (line 70-75)

“An often-used framework to address implementation challenges is the Consolidated Framework for Implementation Research (CFIR). This comprehensive framework describes factors that are important in implementing and evaluating complex interventions. It consists of five domains (characteristics of the intervention, outer setting, inner setting, characteristics of individuals, and process of implementation) which interact and determine effectiveness of implementation together [20].”

Methods (line 83-87)

“The CFIR model was used to structure and organize the data. In the present study, intervention characteristics refers to PCPs’ experiences with applying suicide prevention practices such as exploring suicidal feelings. Outer setting refers to the level of collaboration with MHC and in inner setting we discuss the role of MHNs with regard to suicide prevention. The process of implementation describes how the implementation strategy, SUPRANET, was received by PCPs.”

Results (line 144-331)

see manuscript

Discussion (line 335-344)

“The CFIR was used to structure the data and to gain insights in facilitating and limiting factors of SUPRANET. All but one domain (individual characteristics) from the CFIR were explicitly addressed. Since this study was based on PCPs’ perceptions and experiences, individual characteristics are intertwined with other domains, particular with intervention characteristics. This study found that there were important barriers to the assessment of suicide risk in patients (intervention characteristics) because of PCPs’ self-perceived incompetence, burdensomeness of suicide and their lack of time and workload. The relationship with MHC was an important limiting factor from the outer setting and with regard to inner setting, MHSS had a positive influence on implementation. Although there were some difficulties in the process of implementation, SUPRANET itself was perceived as useful strategy to improve suicide prevention practices. “

---

## [Editor Report · Decision Letter 2]

5 Nov 2020

Engaging Primary Care Professionals in suicide prevention: a qualitative study

PONE-D-20-01479R2

Dear Dr. Elzinga,

We’re pleased to inform you that your manuscript has been judged scientifically suitable for publication and will be formally accepted for publication once it meets all outstanding technical requirements.

Kind regards,

Evan M Kleiman

Academic Editor

PLOS ONE
---

## [Editor Report · Acceptance letter]

9 Nov 2020

PONE-D-20-01479R2 

Engaging primary care professionals in suicide prevention: a qualitative study 

Dear Dr. Elzinga:

I'm pleased to inform you that your manuscript has been deemed suitable for publication in PLOS ONE. Congratulations! Your manuscript is now with our production department. 

Kind regards, 

on behalf of

Dr. Evan M Kleiman 

Academic Editor

PLOS ONE